# THE CLOSE RELATIONSHIP BETWEEN CONTRASTIVE LEARNING AND META-LEARNING

**Renkun Ni**[*]
University of Maryland
rn9zm@cs.umd.edu

**Manli Shu**[*]
University of Maryland
manlis@cs.umd.edu

**Hossein Souri**
Johns Hopkins University
hsouri1@jhu.edu

**Micah Goldblum**
University of Maryland
goldblum@umd.edu

**Tom Goldstein**
University of Maryland
tomg@cs.umd.edu

## ABSTRACT

Contrastive learning has recently taken off as a paradigm for learning from un-labeled data. In this paper, we discuss the close relationship between con-trastive learning and meta-learning under a certain task distribution. We com-plement this observation by showing that established meta-learning methods, such as Prototypical Networks, achieve comparable performance to SimCLR when paired with this task distribution. This relationship can be leveraged by taking established techniques from meta-learning, such as task-based data aug-mentation, and showing that they benefit contrastive learning as well. These tricks also benefit state-of-the-art self-supervised learners without using nega-tive pairs such as BYOL, which achieves 94.6% accuracy on CIFAR-10 using a self-supervised ResNet-18 feature extractor trained with our meta-learning tricks. We conclude that existing advances designed for contrastive learning or meta-learning can be exploited to benefit the other, and it is better for contrastive learning researchers to take lessons from the meta-learning literature (and vice-versa) than to reinvent the wheel. Our Pytorch implementation can be found on: https://github.com/RenkunNi/MetaContrastive

## 1 INTRODUCTION

Self-supervised visual representation learning (SSL) has recently gathered attention due to its ability to learn image features without manual supervision, thus allowing for efficient learning on down-stream tasks such as detection and segmentation (Noroozi & Favaro, 2016; Zhang et al., 2016; Oord et al., 2018; Hjelm et al., 2018; Wu et al., 2018; Gidaris et al., 2018; He et al., 2020; Misra & Maaten, 2020; Tian et al., 2020a; Chen et al., 2020a; Kim et al., 2020; Grill et al., 2020; Caron et al., 2020; Kalantidis et al., 2020; Shen et al., 2020; Li et al., 2020; Zbontar et al., 2021; Chen & He, 2021; Caron et al., 2021). Among SSL approaches, contrastive learning based methods (Chen et al., 2020a; He et al., 2020; Chen et al., 2020b) show particularly strong potential and achieve promising results which are close to those of fully supervised methods on numerous computer vision benchmarks.

These methods rely on applying various data augmentations such as random crops, flips, color distor-tion, and Gaussian blur on the same training sample to create different views of an image. Two such example methods, SimCLR (Chen et al., 2020a) and MoCo (He et al., 2020), involve reducing the distance between features corresponding to positive pairs (different augmented views of the same image), and increasing the distance between features corresponding to negative pairs (augmented views of different images).

Meanwhile, meta-learning is an established popular framework for learning models that quickly adapt to on-the-fly tasks given a small number of examples (Hochreiter et al., 2001; Finn et al., 2017; Nichol et al., 2018; Bertinetto et al., 2018; Lee et al., 2019). The training loop for meta-learners typically involves (i) sampling a random batch of classes and (ii) updating a feature ex-

---

[*]equal contribution

tractor to distinguish between these classes. This procedure mirrors that of contrastive learning, which proceeds by (i) sampling a batch of images and augmenting them to generate classes (each "class" is an image plus all of its views), and (ii) updating a feature extractor to distinguish between these classes. Conceptually, this contrastive learning procedure resembles meta-learning where the training tasks are generated by computing multiple views of individual images.

In this paper, we discuss the close relationship between contrastive learning and meta-learning. Concretely, we show that established meta-learning algorithms, originally designed for few-shot learning, can achieve the same performance as recent contrastive learning algorithms on standard SSL problems when paired with the same data sampling strategy. In addition, we explore techniques, originally designed for meta-learning, that can improve contrastive learning. Specifically, we explore ways by which we can adapt data augmentation strategies inspired by recent work in meta-learning (Su et al., 2020; Ni et al., 2021) to SSL and find that this approach can yield significant performance boosts.

Our contributions can be summarized as follows:

- We formulate a meta-learning based framework for understanding self-supervised learning, and we show that meta-learners can achieve comparable self-supervised performance to contrastive learning methods.

- We propose a meta-specific task augmentation strategy which boosts the performance of self-supervised learning. This data augmentation method generalizes to methods with no negative pairs, such as BYOL (Grill et al., 2020), as well.

## 2    RELATED WORK AND BACKGROUND

### 2.1    META-LEARNING FOR FEW-SHOT LEARNING

Meta-learning algorithms for few-shot learning aim to learn a model that can quickly adapt to new tasks with limited data and generalize to unseen examples. To achieve this, the adaptation and evaluation procedures are both simulated during meta-training. During each episode of meta-learning, we sample a task, $\mathcal{T}_i$, from a distribution of tasks, often corresponding to combinations of training classes formed into classification problems. Each task consists of *support* data $\mathcal{T}_i^s$ and *query* data $\mathcal{T}_i^q$, so that $\mathcal{T}_i = \{\mathcal{T}_i^s, \mathcal{T}_i^q\}$. When applied to few-shot classification, this task is called a $k$-shot, $N$-way classification problem, where $k$ denotes the number of training samples per category in the support data. Then, support data will be used to simulate few-shot training data, while query data will be used to simulate novel testing samples.

A meta-learning model $F$, in this setting, contains a feature extractor and a classification strategy, $\mathcal{A}$. This classification strategy can take various forms, such as adding a linear classifier on top of the feature extractor and fine-tuning either the linear layer or the whole network end-to-end, or this strategy may simply classify samples by selecting the nearest class prototype. Meta-learning training algorithms have an *inner loop* and an *outer loop* in each parameter update. In the inner loop, the model is first fine-tuned on support data $\mathcal{T}_i^s$. Then, in the outer loop, the updated model is used to predict on query data $\mathcal{T}_i^q$, and a loss is minimized with respect to the model's parameters before fine-tuning.[1] Intuitively, we update parameters so that the feature extractor extracts better features for the classification strategy, often resulting in tightly clustered features corresponding to each class (Goldblum et al., 2020). Existing works apply various methods for fine-tuning on support data during the inner loop. In a line of algorithms, such as MAML and Reptile (Finn et al., 2017; Nichol et al., 2018), all the parameters in the model are updated using gradient descent during fine-tuning on support data. Other algorithms, such as MetaOptNet and R2-D2 (Lee et al., 2019; Bertinetto et al., 2018), keep the feature extractor frozen during fine-tuning; MetaOptNet uses SVM, and R2-D2 uses ridge regression on top of the feature extractor. Similarly, metric learning approaches, such as ProtoNet (Snell et al., 2017; Kye et al., 2020), freeze the feature extractor as well, and create class centroids from the support data in the inner loop. In this paper, we primarily focus on the latter algorithms due to their efficiency and performance as well as the similarity of contrastive learning to metric learning.

---

[1] Note that algorithms in the vein of Reptile (Nichol et al., 2018) do not split the tasks into support and query.

## 2.2 SELF-SUPERVISED LEARNING

**Contrastive SSL.** Contrastive methods (Oord et al., 2018; Wu et al., 2018; Tian et al., 2020a; Chen et al., 2020a; He et al., 2020; Chen et al., 2020b) achieve promising performance in self-supervised learning. As mentioned previously, contrastive learning maximizes agreement on different augmented views of the same image (called *positive pairs*) while ensuring disagreement on samples generated by different base images (called *negative pairs*). Given a batch of input images $\mathbf{x}$, $m$ random data augmentations are applied on the same batch, generating a set of training samples $\{\tilde{\mathbf{x}}^i\}_{i=1}^m$. These samples are fed into a backbone network $f(\cdot)$ to obtain the feature representations $\{\mathbf{h}^i\}_{i=1}^m$. Then, a small neural network $g(\cdot)$, usually a non-linear MLP, is applied to project $\{\mathbf{h}^i\}_{i=1}^m$ to the latent representations $\{\mathbf{z}^i\}_{i=1}^m$ in the space where a contrastive loss $l(\cdot)$ is applied, ensuring that latent representations of positive pairs are similar while latent representations of negative pairs are different. In this paper, we mainly focus on this type of self-supervised learning and show its close relation with meta-learning. The batch sampling procedure of contrastive learning can be viewed as sampling a new classification problem with a number of classes equal to the number of base images used to generate augmented views. We will see that this on-the-fly sampling of classification problems closely mirrors a common meta-learning setup.

**Non-Contrastive SSL.** Some non-contrastive methods are generative approaches, such as auto-encoders (Vincent et al., 2008; 2010; Kingma & Welling, 2013), and adversarial learning (Goodfellow et al., 2014), where a distribution is learned over data and a latent embedding. These methods are typically computationally expensive as they require training a learned model which maps latent representations to pixel space. Other non-contrastive methods rely on using heuristic designed pretext tasks (Doersch et al., 2015; Zhang et al., 2016; Noroozi & Favaro, 2016; Gidaris et al., 2018) to learn the representation. More recently, BYOL (Grill et al., 2020) showed that by bootstrapping a target representation prediction, feature representations can be learned without negative pairs. However, BYOL still adopts the data augmentation procedure from contrastive learning, where different augmented views are used as training samples. In section 5, we show that BYOL still benefits from our proposed task augmentation strategy.

**Data Augmentation in Meta-Learning and SSL.** Data augmentations play an essential role in both meta-learning and self-supervised learning. In meta-learning, Su et al. (2020); Liu et al. (2020); Ni et al. (2021) show that proper data augmentation and meta-specific task augmentations dramatically improve few-shot learning performance by expanding the number of classes available for sampling. In self-supervised learning, Tian et al. (2020b) show contrastive learners can find better feature representations when views contain less mutual information. In addition, Kim et al. (2020); Shen et al. (2020); Li et al. (2020) show that adding harder examples such as cut-mixed samples into the training pipeline can improve self-supervised performance. In this paper, we show that similarly to meta-learning, self-supervised learners can benefit from carefully applied data augmentation techniques which mirror task augmentations from the meta-learning literature.

## 2.3 SELF-SUPERVISED LEARNING FOR META-LEARNING AND FEW-SHOT LEARNING

Another line of research (Hsu et al., 2018; Khodadadeh et al., 2018; Ye et al., 2020; Medina et al., 2020) focuses on self-supervised learning for meta-learning and few-shot learning. Hsu et al. (2018) focuses on partitioning samples from a dataset to construct meta-learning tasks and using MAML or ProtoNet on 4-layer architectures to solve few-shot problems. Similarly, Khodadadeh et al. (2018) and Medina et al. (2020) add more data augmentations, such as Auto Aug (Cubuk et al., 2018) and use MAML and ProtoNet, respectively, to learn a few-shot representation. To further improve few-shot performance, Ye et al. (2020) sample harder mixed support examples and apply a task-specific projection head to help generalize to unseen classes. These methods focus on few-shot learning performance, which entails up to 50 training examples per class. In contrast, we focus on the unsupervised learning paradigm where large models are pre-trained on samples generated via data augmentation and are applied to downstream tasks such as ImageNet classification.

## 3 EXPERIMENTAL SETUP

**Datasets and Evaluation** We conduct self-supervised training on both the CIFAR-10 and ImageNet datasets (Krizhevsky et al., 2009; Deng et al., 2009). Following Chen et al. (2020a), we evaluate

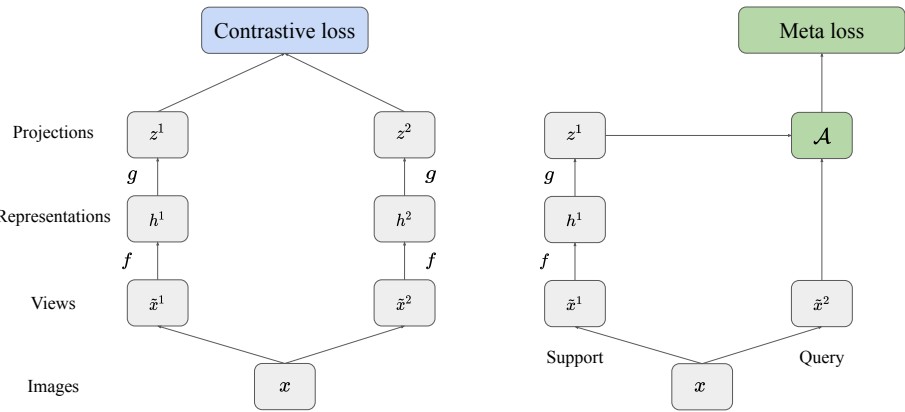

(a) Training procedure of contrastive learning  (b) Meta-Learning framework for SSL

Figure 1: (a) Training procedure of contrastive learning. Two augmented views are generated by applying random transformations to the same input batch. A backbone $f(\cdot)$ and a projector $g(\cdot)$ is learned through contrastive prediction tasks. (b) Meta-Learning framework for SSL. We adopt the same data augmentation operations as contrastive learning. We generate a $b$-way classification problem, $b$ is the batch size, by treating each image itself as a class. Two views are separated as *support* data, on which the network is fine-tuned and the classification strategy is learned by $\mathcal{A}$; *query* data, on which we apply the updated model and calculate the meta-loss. At the end of training, everything but $f(\cdot)$ is discarded, and $\mathbf{h}$ is used as the image representation.

pre-trained representations in a linear evaluation setting, where feature extractors are frozen, and a classification head is stacked on top and tuned. In addition, we test the performance of the pre-trained feature extractors on downstream tasks such as transfer learning and semi-supervised learning with 1% and 10% of labeled data. Evaluation and dataset details can be found in Appendix A.2.

**Pre-Training Details** We use a ResNet-18 backbone for all experiments on CIFAR-10 and ResNet-50 for those on ImageNet. We train the model on CIFAR-10 with the LARS optimizer (You et al., 2019) and batch size 1024 for 1000 epochs (with 4 GPUs). On ImageNet, we use the same optimizer and batch size of 256, and we train for 100 epochs (with 8 GPUs). For ImageNet pre-training, we follow the hyperparameter setting in Chen et al. (2020a), including baseline data augmentation methods, dimension of the latent space, and learning rate decay schedule. For CIFAR-10 pre-training, we use the same CIFAR-10 specific hyperparameters as SimCLR again. For BYOL, we use the same learning rate schedule as meta-learners and start with learning rate 4. In addition, both the projector and predictor in BYOL are two-layer MLPs with hidden dimension 2048 and output dimension 256. More details can be found in Appendix A.1.

## 4  A META-LEARNING FRAMEWORK FOR SSL

Meta-learners, designed for few-shot learning, and contrastive learners for SSL are built on similar intuitions. Both approaches learn to solve new tasks on-the-fly with each batch – new classification problems in the case of meta-learning and differentiating a new batch of images in the case of contrastive learning. Furthermore, both approaches hold a goal of learning invariances which generalize to novel problems at inference; meta-learners should extract similar features for each instance of a novel test class, and contrastive learners should extract similar features for each view of an image sample. In this section, we show how one can construct a meta-learning framework for SSL which closely mirrors the strategy adopted by recent contrastive learning methods.

We now describe how to generate meta-learning task distributions $p(\mathcal{T})$ for self-supervised learning. We adopt the data augmentation operations from contrastive learning, where different random augmentations are applied to the input batch to generate alternative views. In general, given $m$ random augmented views of a batch of $b$ input images $\mathbf{x}$, we can create a $b$-way classification problem by treating all images generated by the same base image as a class. Then, we can divide the data from each class into $m_1$ support and $m_2$ query samples so that $m_1 + m_2 = m$. This framework for

---

**Algorithm 1:** Meta-Learning Framework for Self-Supervised Learning

---

**Require:** Base model $F_\theta$, classification strategy $\mathcal{A}$, learning rate $\gamma$, and distribution of data augmentations $\mathcal{D}$.

Initialize $\theta$, the weights of $F$;

**while** *not done* **do**

    **for** $j = 1, ..., n$ **do**

        Sample a batch of base images $\mathbf{x}$;

        Sample $m$ random data augmentations from $\mathcal{D}$ to obtain augmented views $\{\tilde{\mathbf{x}}^{\mathbf{i}}\}_{i=1}^{m}$;

        Separate $\{\tilde{\mathbf{x}}^{\mathbf{i}}\}_{i=1}^{m}$ into support set $\mathcal{T}_j^s = \{\tilde{\mathbf{x}}^{\mathbf{i}}\}_{i=1}^{m_1}$ and query set $\mathcal{T}_j^q = \{\tilde{\mathbf{x}}^{\mathbf{i}}\}_{i=m_1+1}^{m}$;

        Fine-tune model on $\mathcal{T}_j$: $\theta_j = \mathcal{A}(\theta, \mathcal{T}_j^s)$;

        Compute gradient $g_j = \nabla_\theta \mathcal{L}(F_{\theta_j}, \mathcal{T}_j^q)$;

    **end**

    Update base model parameters: $\theta \leftarrow \theta - \frac{\gamma}{n} \sum_j g_j$.

**end**

---

sampling a batch containing augmented views of base images and dividing them into support and query samples yields a task distribution $p(\mathcal{T})$. In few-shot learning terminology, each training task $\mathcal{T}_i$ is a $m_1$-shot-$b$-way classification problem, since we have $b$ base images which generate classes, and for each such image, we have $m_1$ support samples. Viewed in this way, SimCLR sets $m = 2$ and $m_1 = m_2 = 1$, but SimCLR has important differences from common meta-learners.

Unlike typical meta-learning methods, SimCLR compares every sample with every other sample, while methods like ProtoNet only compare each query sample with each support prototype and not with each other. Moreover, SimCLR samples a single large batch of samples for each episode of training which corresponds to sampling a single task, while meta-learners typically sample a batch of many tasks, i.e., classification problems, during each episode.

Now that we have established a framework for sampling tasks, we can directly apply various meta-learning algorithms, such as R2-D2 and ProtoNet described in Section 2.1, in order to learn the parameters $\theta$ of the base model $F$. Recall that the base model contains a classification strategy and a feature extractor, which is the learning target of SSL. We use the same feature extractor here used for contrastive learning in Section 2.2, which consists of a backbone $f(\cdot)$ followed by a projection head $g(\cdot)$. Formally, we solve the meta-learning optimization problem,

$$\min_\theta \mathbb{E}_\mathcal{T}\Big[\mathcal{L}_{SSL}\Big], \quad \mathcal{L}_{SSL} = l(F_{\theta'}, \mathcal{T}^q),$$

where $\theta' = \mathcal{A}(\theta, \mathcal{T}^s)$ are parameters updated by training on support tasks, and $l$ is the loss function, e.g., cross-entropy loss in our work, used in the outer loop of training. After pre-training, only the backbone $f(\cdot)$ will be kept for self-supervised evaluation. This meta-learning framework for self-supervised learning is summarized in Algorithm 1 and Figure 1.

We compare the performance of representations learned by meta-learners with SimCLR under the default setting described in Section 3. Table 1 and Table 2 show the linear evaluation top-1 accuracy for the feature representations trained and tested on the CIFAR-10 and ImageNet datasets, respectively. We observe that representations learned via meta-learning (R2-D2 and ProtoNet) can achieve performance on par with SimCLR on CIFAR-10 but worse on ImageNet. Note that during training, we use the same exact hyperparameter as SimCLR due to computational constraints, which may

Table 1: Linear evaluation on CIFAR-10 for representations learned via contrastive learning and our meta-learning framework.

Table 2: Linear evaluation on ImageNet for representations learned via contrastive learning and our meta-learning framework.

| Method | Backbone | Top-1 Acc(%) |
|---|---|---|
| SimCLR | ResNet-18 | 91.4 |
| ProtoNet | ResNet-18 | 91.8 |
| R2-D2 | ResNet-18 | 91.6 |

| Method | Backbone | Top-1 Acc(%) |
|---|---|---|
| SimCLR | ResNet-50 | 58.8 |
| ProtoNet | ResNet-50 | 57.6 |
| R2-D2 | ResNet-50 | 55.5 |

Table 3: ImageNet Top-1 accuracy (%) of models fine-tuned with few labels.

| Method | Backbone | Label fraction | |
| --- | --- | --- | --- |
| | | 1% | 10% |
| Supervised baseline | ResNet-50 | 25.4 | 56.4 |
| SimCLR | ResNet-50 | 32.4 | 53.6 |
| ProtoNet | ResNet-50 | 31.0 | 52.9 |
| R2-D2 | ResNet-50 | **37.9** | **58.8** |

not be optimal specifically for our proposed method. We will see in the following experiments that although R2-D2 achieves worse linear evaluation on ImageNet with this hyperparameter setting, it actually performs better than SimCLR on downstream tasks, such as semi-supervised learning and transfer learning, other popular (and plausibly more realistic) evaluation scenarios for SSL methods.

Following Chen et al. (2020a), we first evaluate the pre-trained model by semi-supervised learning, where we fine-tune the pre-trained model with only a fraction of labeled ImageNet data (1% and 10%). As we see from Table 3, with the same fine-tune setting (See Appendix A.2), models pre-trained by R2-D2 can achieve $\sim 5\%$ higher top-1 accuracy than those pre-trained by SimCLR after fine-tuning on labeled data. Notably, the supervised baseline from Zhai et al. (2019) is strong due to exhaustive hyperparameter searching and stronger data augmentations used during the training.

To further compare the feature representations learned by different methods, we apply the pre-trained weights to transfer learning. We consider 8 datasets with natural images of various categories (Nilsback & Zisserman, 2008; Cimpoi et al., 2014; Everingham et al.; Maji et al., 2013; Bossard et al., 2014; Xiao et al., 2010; Krizhevsky et al., 2009). For each dataset, we use the backbone (ResNet-50) pre-trained on ImageNet as an initialization for the feature extractor of the downstream classification model. In contrast to linear evaluation, we fine-tune the entire model on the given dataset for 20,000 iterations with the best hyperparameter setting selected on its validation split. Details of our hyperparameter selection are included in Appendix A.2. All models are pre-trained on ImageNet for 100 epochs. We also include a baseline model provided in Chen et al. (2020a) which does not use pre-trained weight as an initialization. Note that the baseline model is tuned to achieve comparable performance with a larger search space for hyperparameters, and it is trained for a longer duration. From the results in Table 4, we find that R2-D2 initialized model consistently outperforms its contrastive counterpart on all 8 datasets. These results suggest that for the same number of epochs, a model trained with R2-D2 works better as an initialization for downstream tasks than one trained with SimCLR. We speculate that this property is connected to R2-D2's few-shot learning driven design and simulation of adapting to new tasks during its inner loop.

Table 4: Transfer learning using ImageNet pre-trained weights. We report mean per-class accuracy (%) on the Flowers and Aircraft datasets, mean average precision (mAP) on the VOC2007 classification dataset, and Top-1 accuracy on the remaining datasets.

| | Flowers102 | DTD | VOC2007 | Aircraft | Food101 | SUN397 | CIFAR-10 | CIFAR-100 |
| --- | --- | --- | --- | --- | --- | --- | --- | --- |
| Baseline | 92.0 | 64.8 | 67.3 | 85.9 | **86.9** | 53.6 | 95.9 | 80.2 |
| SimCLR | 92.4 | 72.7 | 66.0 | 83.7 | 86.3 | 57.4 | 94.8 | 79.1 |
| ProtoNet | 92.7 | 71.5 | 64.7 | 83.9 | 86.2 | 56.4 | 96.0 | 79.1 |
| R2-D2 | **94.5** | **73.8** | **69.9** | **86.2** | **86.9** | **59.7** | **96.7** | **82.8** |

## 5 BOOSTING SSL WITH META-SPECIFIC AUGMENTATION

Now that we have established a relationship between contrastive learning and meta-learning, we will apply tools developed in the latter discipline to enhance contrastive learners. Previous work has shown that data augmentations such as crops and colorizing play an important role in both contrastive learning and meta-learning. We focus on a particular augmentation strategy from the meta-learning literature, termed *task augmentation*, which aims to expand the number of classes

available for sampling rather than expanding the number of samples per class. Liu et al. (2020); Ni et al. (2021) show empirically that data augmentations work best when applied to carefully chosen parts of the meta-learning batch, and large rotations can only work as a task augmentation, in which rotation by a chosen degree is applied to all images in an entire class, and we then treat them as a new class. Large rotations, and other dramatic transformations used for task augmentation, actually decreases performance when instead applied independently on support and query samples (as a way to increase data within a class) rather than uniformly on an entire class (therefore defining a new class). Augmentations that exhibit this task augmentation behavior are typically those which transform an image so much that its semantic content looks different to a human. By keeping images with very large augmentations in the same class, we may accidentally encourage models to learn overly strong invariances which do not naturally exist in the data.

Table 5: Linear evaluation (Top-1 accuracy (%)) on CIFAR-10 with feature representations learned by SimCLR and BYOL in default setting (see Section 3). Simply adding large rotations to data augmentation hurts the performance of self-supervised learning.

| Rotation | SimCLR | BYOL |
|----------|--------|------|
| No       | 91.4   | 92.1 |
| Yes      | 89.7   | 90.6 |

Although data augmentations such as large rotations have been shown effective for visual pre-training (Feng et al., 2019; Gidaris et al., 2018), how to encode that into the data augmentation framework of contrastive learning remains unclear. In addition, we observe that when applied along with contrastive learning, the phenomenon mentioned above occurs as well. Namely, the same large rotation data augmentation, which can improve the performance of meta-learners via task augmentation, also degrades the performance of contrastive learners when applied to samples independently (instead of uniformly to an entire class). Table 5 shows linear evaluation accuracy on CIFAR-10 when we add this augmentation to the training pipeline of SimCLR. In this experiment, we randomly rotate every augmented view by $\{90°, 180°, 270°\}$ with probability 0.25 each. In Table 5, we see that the accuracy of SimCLR drops by $\sim 2\%$, and this degradation also occurs in self-supervised learning methods without negative pairs such as BYOL, which adopts the same augmentation pipeline as SimCLR. Driven by the observation and insights from the meta-learning literature, we are motivated to apply strong augmentations, such as large rotations, to contrastive learning at the task level so that models can benefit from the additional augmentation without learning overly strong and harmful invariances.

In the literature on meta-learning literature for few-shot classification, large rotations can be used either as a task augmentation or an auxiliary loss (Su et al., 2020; Liu et al., 2020; Gidaris et al., 2019). We adopt both methods into our meta-learning inspired pipeline for self-supervised learning and describe the details below. This procedure is illustrated in Figure 2 and Figure 3.

**Large Rotation as Task Augmentation.** Instead of randomly rotating each training sample independently, we rotate all images from the same class (different augmented views of the same original base image) by the same degree (chosen randomly from $\{0°, 90°, 180°, 270°\}$) in both the support and the query data. In such a way, the number of potential classes is enlarged by 3 times. We combine large rotation augmentation with the basic augmentations used in contrastive learning during the sampling stage and keep other components of the training procedure unchanged.

**Large Rotation as Auxiliary Loss.** In addition to task augmentation, large rotations can be used as an auxiliary prediction task for the original self-supervised problem, where the angle of a rotated image is used as the target label. To this end, we spin the input batch $\mathbf{x}$ by an angle to generate the rotated training samples, $\{\mathbf{x}_d, \mathbf{y}\}$, where $d \in \{0°, 90°, 180°, 270°\}, \mathbf{y} = \{d/90\}$. Then, we stack a 4-way classification head $r$ on top of the shared backbone $f$ and projection head $g$, to predict the angle of the rotations evaluated with cross-entropy loss $l_{LR}$:

$$\mathcal{L}_{LR} = \sum_{(x,y)\in\{\mathbf{x}_d,\mathbf{y}\}} l_{LR}(r(g(f(x))),y). \tag{1}$$

During training, we sum this loss with the original self-supervised loss to achieve the final objective, $\mathcal{L} = \mathcal{L}_{SSL} + \lambda\mathcal{L}_{LR}$, where $\lambda$ is a coefficient that controls the influence of our prediction task. In

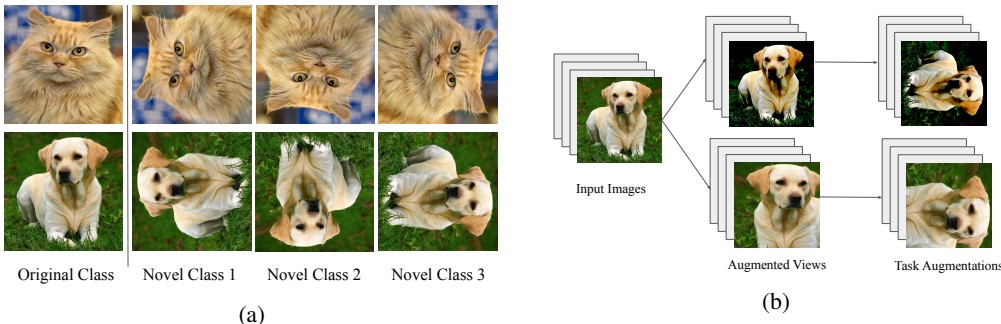

Original Class    Novel Class 1    Novel Class 2    Novel Class 3

(a)

Input Images      Augmented Views    Task Augmentations

(b)

Figure 2: (a) Examples of novel classes created by rotation by $\{90°, 180°, 270°\}$. (b) Adding task augmentation into the data augmentation pipeline. We first apply random transformations on the input batch, then for each augmented view from the same base image, we rotate them by the same degree.

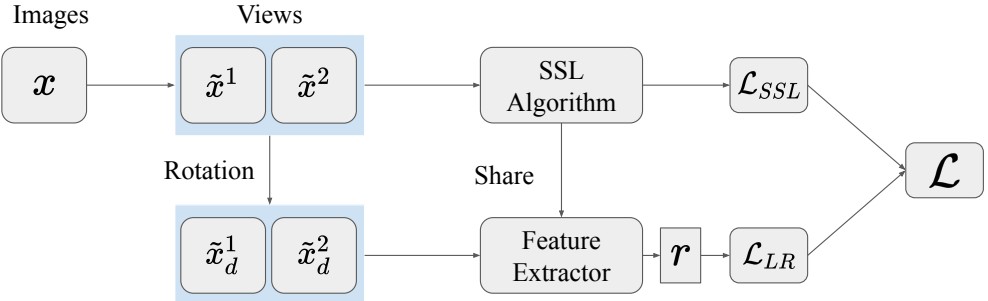

Figure 3: Training procedure with Large Rotation augmentation as an auxiliary loss. We randomly rotate each augmented view by a degree in $\{0°, 90°, 180°, 270°\}$. We share the feature extractor used in the SSL algorithm and apply it along with a 4-way linear head $r(\cdot)$ to predict the rotation angle. We sum up the SSL loss, $\mathcal{L}_{SSL}$, with the angle prediction loss, $\mathcal{L}_{LR}$, as our ultimate objective, $\mathcal{L}$.

contrast to few-shot classification problems, the batch size on self-supervised tasks is often very large. If we rotate all images individually by four different degrees, we will triple the batch size which will consume valuable memory. Instead, we propose to only sample a single rotation for each image. We conduct an ablation study on the number of rotations used to augment each batch on CIFAR-10 pre-raining, and we show that when the batch size is large, rotating each augmented view in the batch once is enough (see Appendix A.3).

Table 6: Linear evaluation (Top-1 accuracy (%) with one standard error) on CIFAR-10 with representations learned via different augmentations."Baseline" denotes augmentations used in SimCLR, "+ TA" denotes adding large rotation as task augmentations, "+ $\mathcal{L}_{LR}$" denotes adding large rotation as an auxiliary loss, and "+ Mix" denotes adding image mixing augmentation (Shen et al., 2020).

| Augmentations | SimCLR | R2-D2 | ProtoNet | BYOL |
|---|---|---|---|---|
| Baseline | $91.52 \pm 0.19$ | $91.46 \pm 0.11$ | $91.84 \pm 0.19$ | $92.08 \pm 0.13$ |
| + TA (ours) | $91.46 \pm 0.13$ | $91.24 \pm 0.06$ | $91.98 \pm 0.13$ | $92.03 \pm 0.15$ |
| + $\mathcal{L}_{LR}$ (ours) | $93.04 \pm 0.17$ | $92.74 \pm 0.11$ | $93.16 \pm 0.13$ | $93.18 \pm 0.16$ |
| + Mix | $92.98 \pm 0.13$ | $93.02 \pm 0.11$ | $93.36 \pm 0.11$ | $93.83 \pm 0.06$ |
| + Mix + $\mathcal{L}_{LR}$ (ours) | $\mathbf{93.88 \pm 0.13}$ | $\mathbf{93.70 \pm 0.20}$ | $\mathbf{94.12 \pm 0.13}$ | $\mathbf{94.57 \pm 0.15}$ |

Table 6 highlights the effectiveness of treating large rotations as a task augmentation or as an auxiliary loss on pre-trained CIFAR-10 representation in the default setting. Notably, our method enables BYOL to achieve $94.6\%$ accuracy with a backbone trained entirely on unlabeled data, just as high

as models of the same architecture trained in a fully supervised setting (but without our augmentations) (Kuang et al., 2017). For task augmentation, we apply large rotations with the probability of 0.25 each on top of the baseline augmentations. For the rotation angle predictor loss, we weight the additional loss term with coefficient $\lambda = 1$ for all experiments except for pre-training with R2-D2, where we set $\lambda = 0.01$. In Table 6, we see that by using large rotations as a task augmentation, we can avoid the massive accuracy drop observed in Table 5. In addition, when we add the angle prediction loss for large rotation as an auxiliary loss, we boost the linear evaluation of all the pre-trained representations by at least $1\%$.

Table 7: Linear evaluation on ImageNet with representations learned by SimCLR and with proposed augmentations.

| Model | Top-1 (%) |
|---|---|
| SimCLR | 58.8 |
| SimCLR + $\mathcal{L}_{LR}$ (ours) | **59.6** |

We further show that our proposed method can be combined with existing data augmentation methods by adding harder examples with mix-up or cut-mix in Table 6. Following Shen et al. (2020), we mix one augmented view from each base image with an augmented view from a second image where the indices used for the second augmented view are reverse ordered. In order to mix images, we randomly apply mix-up or cut-mix with equal probability. Our proposed large rotation prediction loss consistently improves performance on top of the mixing augmentation by $\sim 1\%$ on all pre-training methods, thus leading to more than $2\%$ improvement from the baseline augmentations. On ImageNet, Table 7 displays the top-1 linear evaluation accuracy for methods with and without our proposed rotation angle prediction loss. With the proposed auxiliary loss, we achieve 0.8% improvement on the representation trained for 100 epochs. Additional evaluations for transfer learning and semi-supervised learning can be found in Appendix A.4.

## 6 CONCLUSION

In this work, we discuss the close relationship between contrastive learning and meta-learning. In doing so, we propose a new meta-learning framework for self-supervised learning by converting the contrastive setup into on-the-fly image classification tasks. We show that feature extractors pre-trained via meta-learning achieve comparable results to contrastive learning methods and, in fact, they transfer better to downstream tasks in some cases. In addition, we leverage data augmentation ideas from meta-learning and incorporate them into contrastive learning. We demonstrate that the proposed meta-specific data augmentation consistently improves the performance of popular self-supervised learning algorithms on various datasets.

## ACKNOWLEDGEMENT

Support for this work was provided by the AFOSR MURI program, the Office of Naval Research, the ONR MURI program, the National Science Foundation (DMS-1912866), DARPA GARD (HR00112020007), and the DARPA Young Faculty Award Program. Additional funding was provided by Capital One Bank.

## REPRODUCIBILITY STATEMENT

Our work is fully reproducible. We release our code in the aforementioned github repositories, and we discuss the training details in Section 3 and evaluation details in Appendix A.2.

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

# A    APPENDIX

## A.1    PRE-TRAINING DETAILS

For ImageNet, we follow similar procedure as Chen et al. (2020a) for both SimCLR and our proposed meta-learning algorithms, where we use a batch size 256, a starting learning rate of 0.3 (following LR = 0.3 × BatchSize/256), weight decay $10^{-6}$ and temperature 0.1. We accumulate the gradients and update our model every 8 iterations during the pre-training. For Cifar10, we use a batch size 1024, a starting learning rate of 4. (following LR = 1. × BatchSize/256), weight decay $10^{-6}$ and temperature 0.5.

## A.2    EVALUATION DETAILS

**Linear Evaluation.**    For ImageNet, we follow similar procedure as Chen et al. (2020a), where we use a batch size 4096, a larger learning rate of 1.6 (following LR = 0.1 × BatchSize/256) and longer training of 90 epochs. For CIFAR-10, we train the linear classifier for 200 epochs with batch size 1024, and a linear decay learning rate schedule starts with rate 0.1. We use SGD as the optimizer for linear evaluation on both datasets.

**Semi-supervised Learning.**    Following Chen et al. (2020a), we fine-tune the pre-trained model with 1% and 10% labeled ImageNet data. Due to the computational limit, instead of using batch size 4096, we use the SGD optimizer with a batch size of 256 and momentum of 0.9. We use a linear decay learning rate schedule starts with rate 0.05. We only use the regular data augmentation for ImageNet training (random crop and resize). We do not use any regularization such as weight decay. For both 1% and 10% of the labeled data, we fine-tune for 60 epochs. The supervised baseline results are imported from Zhai et al. (2019), where they trained several thousand models for hyperparameter tuning and using strong data augmentations.

**Transfer Learning.**    We evaluate the performance of models in transfer learning through fine-tuning. We use SSL pre-trained feature extractors with a linear classification head as our model in transfer learning, and we update the entire network during fine-tuning. In this experiment, we use an SGD optimizer with Nesterov momentum (momentum=0.9). We set the batch size to be 256 and train our models for 20,000 iterations. For each dataset, we follow the training/validation/testing split in Chen et al. (2020a) and perform the hyperparameter search on the validation set. We search through different learning rates (ranging from $10^{-2}$ to $10^{-1}$) and weight decays (ranging from $10^{-6}$ to $10^{-3}$, and 0), and choose the best setting by comparing the performance on the validation set. Results in Table 4 are evaluated on the test split, and our models are trained on the union of training and validation set, using the best hyperparameter settings we have found.

## A.3    THE SAMPLE SIZE FOR LARGE ROTATION AUXILIARY LOSS

Table 8 shows the linear evaluation accuracy for representations learned by different self-supervised learning algorithms with different rotation methods. A full rotation method rotates all the augmented views by 3 times, a half rotation method rotates half of the augmented views by 3 times, and the random rotation only rotates all the augmented views by an angle randomly chosen from $\{0°, 90°, 180°, 270°\}$. With different rotation methods, the number of training examples for rotation angle prediction is different. Among all the experiments, we use the same batch size 1024, and the full rotation will generate $1024 \times 2 \times 4$ training examples, the half rotation will generate $512 \times 2 \times 4$ examples and random rotation will only generate $1024 \times 2$ samples, where 2 represents for the number of the random augmentations and 4 is the length of the degree set. From Table 8, we can see that even if we have much fewer training samples when applying random rotation, we can still achieve comparable results. As a result, we use random rotation in all the experiments as it makes the training more memory efficient.

Table 8: Linear evaluation (top-1 accuracy (%)) on CIFAR-10 with representations learned by various SSL methods with different rotation methods.

| Method | SimCLR | R2-D2 | PN |
|---|---|---|---|
| Random Rotation | 93.0 | 92.9 | 93.0 |
| Full Rotation | 92.6 | 92.8 | 92.9 |
| Half Rotation | 92.9 | 92.6 | 92.7 |

## A.4 MORE EVALUATIONS FOR LARGE ROTATION

In this section, we provide additional evaluations for the model pre-trained with our proposed large rotation in Table 9 and Table 10. We conduct the same transfer learning and semi-supervised learning experiments as in Section 4.

Table 9: ImageNet Top-1 accuracy (%) of models fine-tuned with few labels. "+ $\mathcal{L}_{LR}$" denotes adding large rotation as an auxiliary loss during pre-training

| Method | Backbone | Label fraction | |
|---|---|---|---|
| | | 1% | 10% |
| Supervised baseline | ResNet-50 | 25.4 | 56.4 |
| SimCLR | ResNet-50 | 32.4 | 53.6 |
| SimCLR + $\mathcal{L}_{LR}$ | ResNet-50 | 32.0 | 53.5 |

Table 10: Transfer learning using ImageNet pre-trained weights. We report mean per-class accuracy (%) on the Flowers and Aircraft datasets, mean average precision (mAP) on the VOC2007 classification dataset, and Top-1 accuracy on the remaining datasets. "+ $\mathcal{L}_{LR}$" denotes adding large rotation as an auxiliary loss during pre-training

| | Flowers102 | DTD | VOC2007 | Aircraft | Food101 | SUN397 | CIFAR-10 | CIFAR-100 |
|---|---|---|---|---|---|---|---|---|
| Baseline | 92.0 | 64.8 | 67.3 | 85.9 | 86.9 | 53.6 | 95.9 | 80.2 |
| SimCLR | 92.4 | 72.7 | 66.0 | 83.7 | 86.3 | 57.4 | 94.8 | 79.1 |
| SimCLR + $\mathcal{L}_{LR}$ | 93.1 | 73.0 | 64.4 | 85.8 | 86.3 | 57.0 | 96.2 | 82.0 |

## A.5 PRE-TRAINING FOR TABULAR DATASET

Outside of computer vision, contrastive learning pre-training has been shown effective in the tabular data domain as well, especially for fine-tuning with limited data (Somepalli et al., 2021). To further show the effectiveness of meta-learning in this setting, we pre-train on tabular data with the R2D2 head and evaluate on semi-supervised tasks given only 50 training samples. We apply our methods to 5 datasets from OpenML (Vanschoren et al., 2014), and we average over 5 runs with random train-test split on each dataset. Details of the selected datasets can be found in Somepalli et al. (2021). For datasets with binary classification tasks, we evaluate the algorithm with mean AUROC scores, and for datasets with multi-class classification tasks, we evaluate the methods with mean accuracy. Table 11 shows that our meta-learning based pre-training method can indeed achieve comparable accuracy to state-of-the-art contrastive learning methods on tabular data.

Table 11: Semi-supervised evaluation with 50 labeled training samples on 5 tabular datasets.

| Dataset | Pre-train Nethods | Mean Score | Std |
|---|---|---|---|
| adult | - | 84.46 | 1.82 |
| | Contrastive | 86.66 | 1.62 |
| | R2-D2 | 84.26 | 4.34 |
| arrhythmia | - | 72.36 | 6.20 |
| | Contrastive | 75.94 | 4.32 |
| | R2-D2 | 75.64 | 3.26 |
| telco-customer-churn | - | 79.76 | 2.22 |
| | Contrastive | 80.34 | 1.30 |
| | R2-D2 | 81.38 | 1.69 |
| eucalyptus | - | 45.06 | 3.68 |
| | Contrastive | 46.8 | 8.76 |
| | R2-D2 | 49.17 | 9.03 |
| volkert | - | 40.80 | 2.57 |
| | Contrastive | 39.22 | 2.07 |
| | R2-D2 | 40.44 | 1.76 |

