# OpenReview forum: "The Close Relationship Between Contrastive Learning and Meta-Learning"
_ICLR.cc/2022/Conference — ICLR 2022 Poster_

### Official Review · Reviewer_AV99 · 2021-10-29

**Correctness:** 4
**Technical Novelty And Significance:** 3
**Empirical Novelty And Significance:** 3
**Recommendation:** 6
**Confidence:** 4

**Main Review:**

1. The idea of integrating self-supervised learning into meta-learning is novel. The insight of treating different augmentation as task augmentation is conceptually interesting. Even though the paper does not extend such a framework to the level of other meta-learning literature that datasets and tasks are largely different, I believe the concept proposed in the paper may be a good inspiration for future self-supervised learning research.

2. The first empirical idea proposed in the paper is exactly just rotNet (arXiv:1803.07728), and combining it with the joint-embedding approach is also not novel ("Self-Supervised Representation Learning by Rotation Feature Decoupling" CVPR'19). Therefore, it has limited novelty. I do not see how this setting is different when fitting into the meta-learning framework. It's still working as an auxiliary loss.

3. The second empirical idea seems new to the self-supervised learning framework. Even though the idea itself comes from meta-learning is not novel, I think that demonstrating such an idea can work in a self-supervised learning framework is interesting and can probably benefit the general self-supervised learning research community.

4. The authors provided all experimental details for reproducing the results. I understand that it's not possible to reproduce standard SimCLR due to computational budget but it seems that the comparison is not satisfactory. I do not see any hyperparameter search for the baseline SimCLR/BYOL model. These hyperparameters may not be optimal. For example, using a large learning rate of 4 and a high temperature of 0.5 may not be best for training small batch sizes such as 256. Also, all experiments improvements are incremental e.g. Table 6, 7, 8 only demonstrates +1%. Without an exhaustive hyperparameter search, this +1 % is not convincing enough.

5. Overall, the paper is well written.

**Summary Of The Paper:**

This paper proposed a framework to integrate contrastive/self-supervised learning into meta-learning literature. The authors demonstrate that contrastive learning principles implemented in meta-learning methods such as R2-D2 can achieve comparable results on various computer vision tasks. Next, the authors proposed two tricks in meta-learning literature to improve SSL models, including (1) rotation prediction (2) batch gradient accumulation. Both methods demonstrate incremental improvement over the standard SimCLR baseline.

**Summary Of The Review:**

The concept of combining self-supervised learning and meta-learning is interesting and may have a bigger impact in the future. The first trick of rotation prediction has limited novelty. But the second trick is interesting and demonstrated to be useful. Empirical results are all incremental and not convincing enough.

---

> ### Author Response · Authors · 2021-11-23
> **Thanks for your feedback!**
>
> Thanks for your careful review and helpful suggestions.
>
> Our goal when proposing the rotation auxiliary loss is to emphasize that popular contrastive learning algorithms, such as SimCLR, require the same considerations as meta-learners when incorporating aggressive data augmentations.  We agree that the same tool has been applied to other self-supervised learners, and we have updated the section to contextualize our experiments with respect to the work you pointed out.
>
>
> **"Do not see any hyperparameter search for the baseline SimCLR/BYOL model."**
>
> The original SimCLR paper reported results with a range of different batch sizes.  The hyper-parameters we used for batch size 256 (lr=0.3, temp=0.1) on ImageNet are the same as those used for batch size 256 in the original SimCLR paper. For CIFAR-10 experiments we use a larger learning rate of 4 and a high temperature of 0.5 for a batch size of 1024, a hyperparameter combination also adopted directly from the SimCLR paper. Thus, we actually use the batch size and hyperparameter combinations in the same way as the original work. We have now made our hyperparameter choices more clear in the draft.

---

> > ### Comment · Reviewer_AV99 · 2021-11-29
> > **Response to authors**
> >
> > I have read the response from the authors. They have successfully addressed my concerns.
> > I will keep my score and suggest weak acceptance.

---

### Official Review · Reviewer_KYMy · 2021-11-01

**Correctness:** 1
**Technical Novelty And Significance:** 3
**Empirical Novelty And Significance:** 3
**Recommendation:** 5
**Confidence:** 4

**Main Review:**

Strengths
1. Learning self-supervised visual representations via meta-learning is new and interesting.
2. The presented meta-learning framework for self-supervised learning improves over the SimCLR in downstream tasks. This shows the potential of using meta-learning for self-supervised visual representation learning.
3. The tools from meta-learning are proved to benefit SimCLR.

Weaknesses
1. A core claim and also the title of the paper is "contrastive learning is just meta-learning": contrastive learning can be interpreted as a special case of meta-learning with a certain task distribution. However, its main arguments are not convincing to support this statement. In particular, the paper only discusses similar features (and also differences) shared by contrastive learning and meta-learning, such as solving new tasks on-the-fly with each batch and learning invariances which generalize to novel problems at inference. These shared features are not unique or characteristics to either meta-learning or contrastive learning. For example, deep metric learning via a triplet loss also solves new tasks (different anchors, positive/negative examples in each batch) and aims to learn features generalize to new problems (e.g., new faces for face recognition). To claim "contrastive learning is just meta-learning", one should present a general meta-learning framework and show how the contrastive learning framework can fit in that framework as a special case. Also, one should think about the "core characteristics" of these two frameworks, and show one is a subset of the other. For example, I did not see anything "meta" in SimCLR. Otherwise (only showing some shared features), it is similar to proving that "elephants are just mice" because they both have four legs, a head and a tail.

2. The evaluation is inconsistent in different parts of the paper. Section 4 presents results from both linear protocol and downstream tasks while Section 5 and 6 only present the former. Some experiments are only on CIFAR-10 while some are only on ImageNet. It is unclear whether the data augmentation and gradient accumulation help meta-learning or SimCLR in downstream tasks.

3. The idea of Large Rotation as Auxiliary Loss reads similar to prior work on classifying random image rotations as a pretext task for self-supervised learning. What is the main difference?

4. The proposed meta-learning framework for self-supervised learning contains iterations in the inner loop. Prior self-supervised work indicates the number of training iterations/epochs can significantly affect the performance. An experiment is needed to the performance of meta-learning and SimCLR under different epochs/iterations. It is also interesting to see the extra training time added by the inner loop.

5. Is the loss l on page 5 a contrastive loss?


**Summary Of The Paper:**

The paper first shows that the current popular contrastive learning for self-supervised visual representation learning shares some similarity with a meta-learning framework for few-shot learning. This inspires the authors to propose a meta-learning framework for self-supervised learning. In addition, the paper also shows that tools (data augmentation and gradient accumulation) developed in meta-learning can help enhance contrastive learners.

Experimental results show that the proposed meta-learning framework for self-supervised learning outperforms SimCLR, a state-of-the-art contrastive learning method, on multiple downstream tasks in a semi-supervised learning framework, though it performs not as well as SimCLR under the linear evaluation protocol on ImageNet.  Other results show tools developed in meta-learning can help enhance contrastive learners under the linear evaluation protocol.

**Summary Of The Review:**

Learning self-supervised visual representations via meta-learning is new and interesting. Experimental results also indicate its potential effectiveness. But the main claim "contrastive learning is just meta-learning" is not well supported, as detailed in Main Review. The choice of datasets and tasks in different experiments is heuristic, making the results less convincing. Experiments on the impact of iterations within the inner loop are needed to show more in-depth comparison between the meta-learning framework and SimCLR.

---

> ### Author Response · Authors · 2021-11-23
> **Thanks for your feedback!**
>
> Thanks for your careful review and helpful suggestions. Below are detailed responses to each of your comments.
>
> - **"the title of the paper is 'contrastive learning is just meta-learning'."**
>
> We agree that the original title of the paper is misleading, and we emphasize that the purpose of our paper is not to prove that “contrastive learning is just meta-learning,” but rather to explore the close relationship between these fields and to show how knowledge from one field can benefit the other.   We think that the original title of the paper did not accurately represent the topic of our paper, and so we have updated the title to “The Close Relationship Between Contrastive Learning and Meta-Learning” to prevent this confusion.   We have also updated our draft both to relax our language and also to provide a clearer explanation.
>
> The overlap between meta-learning and contrastive learning is not just incidental; knowledge from the meta-learning literature (that is likely to be overlooked by many contrastive learning practitioners) can be adapted to improve performance on contrastive learning tasks.  We feel that it is better for contrastive learning researchers to take lessons from the meta-learning literature (and vice-versa) than to reinvent the wheel.
>
>
> - **"The evaluation is inconsistent in different parts of the paper."**
>
> We initially left out some experiments because of the prohibitive computational costs of running all combinations, but we agree that these experiments are important. Thus, we have now added the following experiments: 1) downstream tasks for protonet 2) downstream tasks for models trained with rotation and gradient accumulation.
>
>
> - **"The idea of Large Rotation as Auxiliary Loss reads similar to prior work on classifying random image rotations as a pretext task."**
>
> Our goal when proposing the rotation auxiliary loss is to emphasize that popular contrastive learning algorithms, such as SimCLR, require the same considerations as meta-learners when incorporating aggressive data augmentations.  We agree that the same tool has been applied to other self-supervised learners, and we have updated the section to contextualize our experiments with respect to the work you pointed out.
>
>
> - **"The proposed meta-learning framework for self-supervised learning contains iterations in the inner loop."**
>
> We use meta-learners, such as R2-D2 and ProtoNet, which have a closed-form solution in the inner loop and do not involve iterative gradient updates as these are significantly faster than MAML-style algorithms and are also effective.  Thus, the meta-learners do not require extra iterations compared to SimCLR for the same number of epochs.  The style of meta-learner we use, which fixes the feature extractor in the inner loop and also includes MetaOptNet and MCT, has become popular in the last few years and includes many of the highest performing meta-learners for few-shot classification. Also, in all experiments, we only sample one training task during each iteration which ensures that our total training iterations are the same as the baselines.
>
>
> - **"Is the loss l on page 5 a contrastive loss?"**
>
> We use cross-entropy as our loss function in the outer loop and have updated our paper to clarify.
>
> Do you have any additional thoughts we can address during the discussion period?

---

### Official Review · Reviewer_AjN4 · 2021-11-02

**Correctness:** 3
**Technical Novelty And Significance:** 2
**Empirical Novelty And Significance:** 2
**Recommendation:** 5
**Confidence:** 3

**Main Review:**

Strengths:
1. Mathematical formalization of the connection between contrastive learning and meta-learning.
2. New data augmentation techniques for contrastive learning lead to improvements on standard evaluations.
3. Paper is clear and easy-to-read.
4. Code is provided in the supplement.

Weaknesses:
1. The main weakness with the paper is the lack of comparison to the large body of work comparing unsupervised learning and meta-learning. Most notably, Hsu et al. (2019) showed that meta-learning algorithms can be used for unsupervised learning, and many of the papers that cite it have explored transferring techniques in both directions. The novelty here seems to be just the connection to contrastive learning specifically.
2. It is unclear if any of the methods proposed here would extend to domains such as text, which is not examined.
3. [minor] The paper title is aggressive, suggesting that we can stop doing contrastive learning and focus on meta-learning, even though the only connection is in the data generation during training, not in the goals of the two learning paradigms.

Comments:
1. [*The training loop for meta-learners typically involves (i) sampling a random batch of classes and (ii) updating a feature extractor to distinguish between these classes.*] This describes the few-shot learning subset of meta-learning; meta-learning itself is much broader, including applications to supervised learning and RL that do not involve sampling classes.
2. [*In the inner loop, the model is first fine-tuned on support data T_s_i . Then, in the outer loop, the updated model is used to predict on query data T_q_i , and a loss is minimized with respect to the model’s parameters before fine-tuning.*] Algorithms such as Reptile do not separate task data into support and query data.
3. [*These methods usually require operations in pixel space, which is computationally expensive.*] What does this mean? Don’t most deep nets apply at least one operation to pixel space?

References:
Hsu, Finn, Levine. *Unsupervised Learning via Meta-Learning*. ICLR 2019.

**Summary Of The Paper:**

This paper formalizes a connection between the training procedures of (few-shot) meta-learning and contrastive learning. It shows that meta-learning algorithms can be used to pretrain on image data and outperform standard contrastive learning on downstream tasks. The authors further use the connection to develop data augmentation procedures for contrastive learning.

**Summary Of The Review:**

The main concern with this work is that the main insight seems to be a somewhat more specific variant of observations made in past work on the connection between modern unsupervised and meta-learning. While there are some interesting experimental results, I lean against accepting given the lack of any analysis concerning what the novelty is here compared to those papers.

---

> ### Author Response · Authors · 2021-11-23
> **Thanks for your feedback!**
>
> Thanks for your careful review and helpful suggestions. Prompted by your comments, we have updated the related work section to make it more specific and clear. We also added an experiment to show the effectiveness of our methods on tabular data outside of the image domain. Below are detailed responses to each of your comments.
>
> - **"The main weakness of the paper is the lack of comparison to the large body of work comparing unsupervised learning and meta-learning."**
>
> Thanks for pointing out this related work, and we agree that it is highly relevant.  We have added a discussion concerning this line of work to our Related Work section.  [1] Hsu et al. (2018) focuses on partitioning samples from a dataset to construct meta-learning tasks and using MAML or ProtoNet on 4-layer architectures to solve few-shot problems (“up to 50 training examples per class”). In contrast, we focus on the contrastive learning paradigm where larger models are pre-trained on samples generated via data augmentation and are applied to downstream tasks such as ImageNet classification.  Other works in the vein of [1] Hsu et al. (2018) also focus primarily on the few-shot setting [2,3,4]. Please refer to Section 2.3 in our updated draft.
>
> - **"It is unclear if any of the methods proposed here would extend to domains such as text, which is not examined."**
>
> Thanks for your suggestion that we expand our work outside of image classification.  Prompted by your review, we have now tried our method in the tabular data domain, where contrastive learning pre-training has recently been shown effective, especially for fine-tuning on limited labeled data (see [5] Someppali et al. (2021)). We pre-train on tabular data with an R2D2 head and evaluate on semi-supervised tasks given only 50 training samples. We apply our methods to 5 datasets, and we produce 5 runs with random train-test split on each dataset. The table in Appendix A.5 shows that our meta-learning method indeed achieves comparable accuracy to state-of-the-art contrastive learning methods for tabular data.  We have added a section to the appendix of our draft and will move this to the main body after running additional experiments.  We also want to point out that we originally chose to focus on the image domain in which contrastive learning with positive samples generated via data augmentation has become popular, whereas the most common form of pre-training for language is MLM.
>
> - **"The paper title is aggressive"**
>
> Thanks for pointing this out. We have now updated our title to “The Close Relationship Between Contrastive Learning and Meta-Learning”.  We agree that the title was too provocative and aggressive, and more importantly, did not accurately convey the purpose of the paper.  Our message in the paper is that there is a close relationship between contrastive learning and meta-learning which can benefit practitioners, and we do not suggest that practitioners stop doing contrastive learning.
>
> - **Regarding the comments**
>
> We have now updated our draft to clarify the class of meta-learning algorithms we consider.  While the term “meta-learning” is quite broad, we focus our discussion on a (large and mainstream) subset of meta-learning algorithms that certainly does not characterize all work in this space. In addition, we make it more clear why generative methods are typically more computationally expensive since they learn an additional generator mapping the latent representations to the pixel space.
>
> Do you have any additional thoughts we can address during the discussion period?
>
> **References**:
>
> [1] Hsu K, Levine S, Finn C. Unsupervised learning via meta-learning[J]. arXiv preprint arXiv:1810.02334, 2018.
>
> [2] Khodadadeh S, Bölöni L, Shah M. Unsupervised meta-learning for few-shot image classification[J]. arXiv preprint arXiv:1811.11819, 2018.
>
> [3] Ye H J, Han L, Zhan D C. Revisiting Unsupervised Meta-Learning: Amplifying or Compensating for the Characteristics of Few-Shot Tasks[J]. arXiv preprint arXiv:2011.14663, 2020.
>
> [4] Medina C, Devos A, Grossglauser M. Self-supervised prototypical transfer learning for few-shot classification[J]. arXiv preprint arXiv:2006.11325, 2020.
>
> [5] Somepalli G, Goldblum M, Schwarzschild A, et al. SAINT: Improved Neural Networks for Tabular Data via Row Attention and Contrastive Pre-Training[J]. arXiv preprint arXiv:2106.01342, 2021.

---

### Official Review · Reviewer_vRLC · 2021-11-02

**Correctness:** 3
**Technical Novelty And Significance:** 2
**Empirical Novelty And Significance:** 3
**Recommendation:** 6
**Confidence:** 3

**Main Review:**

Overall the paper is quite well written and easy to understand and follow. As said above, novelty might be a bit limited but I can see the benefit of highlighting the relationship of meta- and contrastive learning for many readers in the community. Experimental results are in favor of the approach, however a more compressive analysis (e.g., confidence intervals) would have been appreciated.

**Summary Of The Paper:**

The paper at hand shows relations from self-supervised-learning (e.g., contrastive learning) and meta-learning (e.g., fine-tuning). Whereas the shown relationship is interesting it's not that novel and I would even argue that several teams have implemented similar ideas already. However it has not made that explicit, especially the gradient accumulation for contrastive learning which is quite relevant for many  learning in real world applications.

**Summary Of The Review:**

A good summary of the state-of-the-art and highlighting some interesting relationships.

---

> ### Author Response · Authors · 2021-11-23
> **Thanks for your review!**
>
> Thanks for your careful review and helpful suggestions. Following your suggestions, we have added confidence intervals for CIFAR-10 results (Table 6 in our draft). We will add confidence intervals for ImageNet experiments in the camera-ready version, but due to high computational costs, we are still waiting for these runs to complete.

---

### Public Comment · ~Haoqing_Wang1 · 2021-11-19
**The same idea has been studied in unsupervised few-shot learning**

The idea of combining a large batch size and multiple data augmentations was first proposed in [1] to solve unsupervised few-shot learning, which is very similar with unsupervised representation learning. In [2], the authors verified that this method has better representation learning capabilities than SimCLR and MoCo-v2 on mini-ImageNet.

[1] Self-supervised prototypical transfer learning for few-shot classification.     ICML 2020 workshop.
[2] Revisiting Unsupervised Meta-Learning: Amplifying or Compensating for the Characteristics of Few-Shot Tasks.   arXiv preprint

---

> ### Author Response · Authors · 2021-11-23
> **Thanks for pointing out the references!**
>
> Thanks for pointing out these additional few-shot learning references!  We have updated our draft with a discussion surrounding this line of work.

---

### Decision · Program_Chairs · 2022-01-20

**Decision:**

Accept (Poster)

**Comment:**

This paper was borderline, based on the reviews. The paper points out an interesting connection (somewhat known but not in this specific version) and good experimental results. However, numerous reviewers raised concerns that the paper was lacking a comparison to prior work connecting unsupervised learning and meta-learning, most notably, Hsu et al. (2019).

After reading the revised version of the paper, the authors address this issue and also all the other reviewer comments. In relation to prior work they clarify that they focus on the contrastive unsupervised case and also do a good job in answering other reviewer concerns relative to novelty and results.

I would also like to point out, as reviewers also did that the previous title was a bit aggressive and provocative. Gladly the authors agree to change it to a more scientific `The Close Relationship Between Contrastive Learning and Meta-Learning”.

Overall I think the authors have done a good effort on addressing the reviewer concerns and I think the paper would be interesting for ICLR readers.